# DeLiVR: Differential Spatiotemporal Lie Bias for Efficient Video Deraining

**Shuning Sun**[1]* **Jialang Lu**[3]* **Xiang Chen**[4] **Jichao Wang**[1]
Dianjie Lu[5] Guijuan Zhang[5] Guangwei Gao[6] Zhuoran Zheng[2]†
[1]University of Chinese Academy of Sciences [2]Qilu University of Technology
[3]Hubei University [4]Nanjing University of Science and Technology
[5]Shandong Normal University [6]Nanjing University of Posts and Telecommunications
zhengzr@njust.edu.cn

## ABSTRACT

Videos captured in the wild often suffer from rain streaks, blur, and noise. In addition, even slight changes in camera pose can amplify cross-frame mismatches and temporal artifacts. Existing methods rely on optical flow or heuristic alignment, which are computationally expensive and less robust. To address these challenges, Lie groups provide a principled way to represent continuous geometric transformations, making them well-suited for enforcing spatial and temporal consistency in video modeling. Building on this insight, we propose DeLiVR, an efficient video deraining method that injects spatiotemporal Lie-group differential biases directly into attention scores of the network. Specifically, the method introduces two complementary components. First, a rotation-bounded Lie relative bias predicts the in-plane angle of each frame using a compact prediction module, where normalized coordinates are rotated and compared with base coordinates to achieve geometry-consistent alignment before feature aggregation. Second, a differential group displacement computes angular differences between adjacent frames to estimate a velocity. This bias computation combines temporal decay and attention masks to focus on inter-frame relationships while precisely matching the direction of rain streaks. Extensive experimental results demonstrate the effectiveness of our method on publicly available benchmarks. The code is publicly available at https://github.com/Shuning0312/ICLR-DeLiVR.

## 1 INTRODUCTION

With the rapid development of mobile terminals and video platforms, outdoor video data inevitably suffers from various adverse weather conditions, among which rain is the most common degradation (Yang et al., 2017; Bahnsen & Moeslund, 2019). Raindrops and rain streaks not only severely reduce the visual quality of videos, leading to blurred details and decreased contrast, but more importantly, they significantly affect the performance of downstream high-level computer vision tasks, such as object detection, semantic segmentation, and scene understanding in autonomous driving (Dai et al., 2021; Singha & Bhowmik, 2020). Therefore, developing efficient and robust video deraining algorithms is crucial for improving the accuracy of advanced vision tasks such as autonomous driving and robotic navigation.

Early video deraining methods mainly relied on hand-crafted priors, such as applying frequency-domain filtering based on the physical characteristics of rain streaks, or using sparse coding and Gaussian mixture models for layered modeling (Li et al., 2018). However, these previous models were overly simple and often suffered from weak generalization when faced with diverse and dynamic real rain scenes, which could easily introduce artifacts or lead to oversmoothed details (Wang et al., 2022). Later, deep learning-based methods, especially CNN, achieved significant improvements through end-to-end learning (Yu et al., 2021; Wang et al., 2020; 2025). Nevertheless, their

---

*Equal contribution.
†Corresponding author.

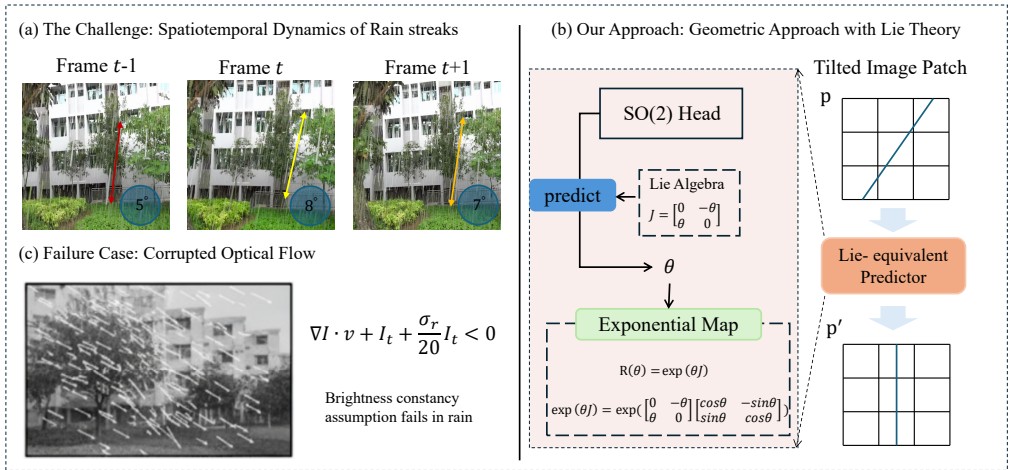

Figure 1: Illustration of the problem and our solution. (a) Challenge: rain streaks show spatiotemporal dynamics with varying angles, making alignment unreliable. (b) Our approach: an SO(2) Head with exponential map ensures geometry-consistent alignment. (c) Failure case: optical flow is corrupted as brightness constancy breaks in rain.

exploitation of temporal information remained limited, mostly through simple frame fusion or optical flow alignment, making it difficult to effectively handle large-scale motion (Guo et al., 2023).

In recent years, some studies have attempted to introduce more powerful architectures to improve video deraining performance, such as adopting Transformers to capture long-range spatiotemporal dependencies (Li et al., 2024; Yan et al., 2021; Yang et al., 2022; Wu et al., 2024). However, these methods lack knowledge of geometric alignment and are overly reliant on data-driven knowledge when utilizing inter-frame information. When training data contains complex kinematic features such as rotations and rapid movements, these methods often exhibit biased information capture under varying rain densities and slight variations in camera pose (Wang et al., 2019; Chan et al., 2022). How to explicitly introduce physically meaningful kinetic knowledge into the network to guide precise video-based rain removal remains a key challenge (Rota et al., 2023; Zhou et al., 2022). To address this issue, we propose to explicitly encode geometric priors of the Lie group (Lie & Engel, 1893) into the attention mechanism, enabling the network to leverage physically interpretable motion constraints to distinguish true correspondences from rain noise during cross-frame aggregation (see Figure 4). This design provides explicit geometric guidance for self-attention, thereby significantly improving alignment robustness under complex motion and subtle camera pose variations.

Specifically, we propose an efficient video de-raining method named DeLiVR, which introduces a novel differential spatio-temporal Lie Bias to effectively estimate the variation of rain streaks in dynamic scenes. Unlike existing methods that rely on unreliable optical flow or unconstrained implicit learning, DeLiVR incorporates continuous geometric transformation theory (Lie groups) as a strong prior, directly injected into the attention mechanism. Our method contains two complementary components. First, we design a rotation-bounded Lie relative bias module, which employs a compact prediction network to directly estimate the in-plane rotation angle of each frame, and achieves geometry-consistent coordinate alignment under the Lie-group framework. Second, we introduce differential group displacement to compute angular differences between adjacent frames, thereby estimating angular velocity and providing dynamic information about motion trends. These two biases are ultimately integrated into a unified attention bias, combined with temporal decay and an attention mask, guiding the network to estimate the intensity and direction of rain streaks.

The main contributions of this paper are summarized as follows:

- We introduce Lie group theory into video deraining for the first time and propose a novel differential spatiotemporal Lie mechanism. This mechanism provides a new paradigm based on geometric priors, independent of optical flow, for solving the feature alignment problem in dynamic scenes.

- We design two bias components: rotation-bounded Lie relative bias and differential group displacement. These components explicitly encode inter-frame rotation transformations and angular velocity information into attention, greatly boosting the spatio-temporal modeling capability of networks in complex rainy videos.

- We conduct extensive experiments on multiple synthetic and real-world deraining benchmarks. The results demonstrate that DeLiVR outperforms existing advanced methods by producing clearer details, more thorough rain removal, and stronger temporal consistency. Moreover, our approach achieves a certain degree of improvement in accuracy for advanced visual tasks.

## 2 RELATED WORKS

**Video Deraining.** Video deraining seeks to recover clear frames from rainy videos where streaks and veiling effects obscure details and introduce spatiotemporal artifacts. Early approaches relied on handcrafted priors such as photometric modeling, frequency-domain filtering, or sparse/low-rank decomposition, which worked in simple cases but easily failed under dense rain or fast motion (Wang et al., 2024; Kim et al., 2015; Li et al., 2018). With the rise of deep learning, CNN-, RNN-, and GAN-based models have achieved better restoration by learning semantic and structural cues, yet frame-wise processing and optical-flow alignment remain computationally expensive and fragile, often leading to mismatches and temporal jitter (Wang et al., 2019). More recently, Transformers and diffusion models have exploited non-local dependencies and generative priors to improve detail recovery, but without explicit geometric constraints, attention struggles to maintain alignment under camera rotations or shake (Liang et al., 2022; Gao et al., 2025). To address this gap, we develop *spatiotemporal Lie-group differential biases* directly into the attention score domain, combining geometry-consistent relative biases with temporal motion cues, which enhances alignment and stability without relying on fragile optical flow estimation.

**Lie-group research focuses on computer vision.** Video restoration critically relies on accurate temporal alignment, yet optical-flow-based methods often fail under rain and other degradations, while implicit attention-based approaches improve robustness but lack interpretability (Wang et al., 2019; Chan et al., 2022). To enhance stability, recent studies incorporate geometric priors into Transformers: from absolute and relative positional encodings (Liu et al., 2021; Su et al., 2024) to Lie-group formulations that model continuous symmetries (Hutchinson et al., 2021). However, strict equivariant designs are computationally demanding and struggle with temporal dynamics. Building on these insights, our method injects lightweight *Lie-group differential biases* into attention, combining frame-level rotation priors with temporal displacements to achieve geometry-consistent and efficient alignment in rainy videos.

## 3 METHODOLOGY

### 3.1 OVERVIEW

As illustrated in Fig. 2, **DeLiVR** restores clean video frames by injecting *Lie-group spatiotemporal biases* into the Transformer backbone. Specifically, each input clip is first divided into patches and embedded into tokens, followed by a lightweight SO(2) head that predicts per-frame rotations to capture camera pose variations. These rotations are then used to build two complementary priors: a spatial bias that enforces geometry-consistent alignment across frames, and a temporal bias that reflects relative angular displacements. The two biases are fused with temporal decay and masking strategies to form a unified spatiotemporal bias, which is directly added to the self-attention process. Guided by this bias, the Transformer backbone focuses on reliable spatial-temporal correspondences, and the attended features are finally decoded to reconstruct the clean video frame.

### 3.2 SO(2) HEAD

To capture frame-wise orientation variations in rainy videos, we design an SO(2) head that predicts in-plane rotations and provides geometry-aware priors for subsequent bias construction.

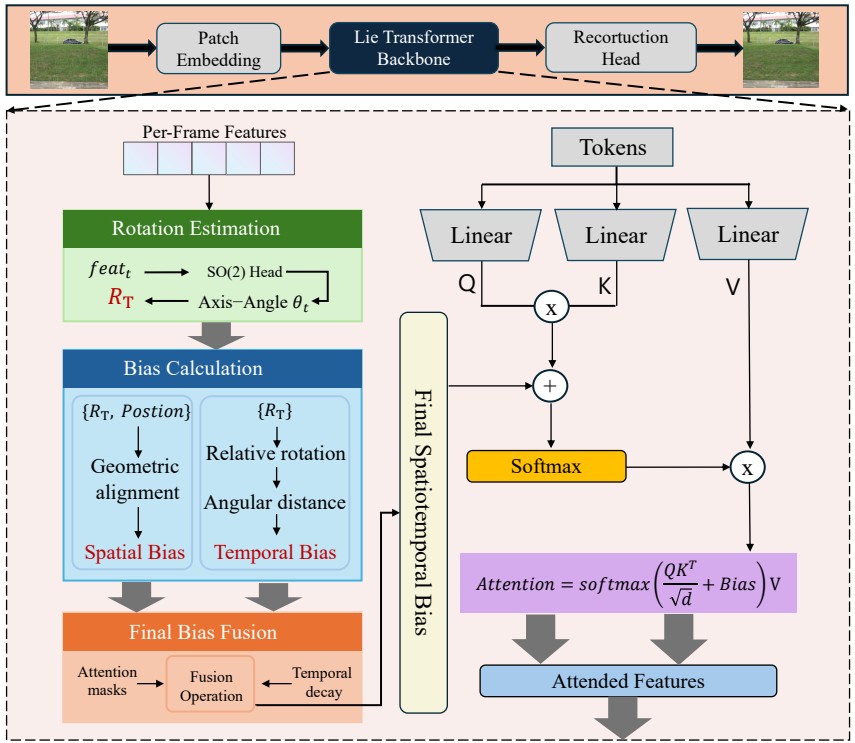

Figure 2: Overall architecture of **DeLiVR**. The model restores clean video frames by estimating per-frame rotations, constructing spatial and temporal biases, and injecting them into biased self-attention for robust geometry-consistent and temporally reliable restoration.

**Preliminaries on SO(2).**    The group SO(2) denotes the set of $2 \times 2$ rotation matrices preserving orientation in the plane:

$$\text{SO(2)} = \left\{ R(\theta) = \begin{bmatrix} \cos\theta & -\sin\theta \\ \sin\theta & \cos\theta \end{bmatrix} : \theta \in \mathbb{R} \right\}. \tag{1}$$

Lie algebra $\mathfrak{so}(2)$ consists of all $2 \times 2$ skew-symmetric matrices, parameterized by a scalar angle $\theta$:

$$\mathfrak{so}(2) = \left\{ \begin{bmatrix} 0 & -\theta \\ \theta & 0 \end{bmatrix} : \theta \in \mathbb{R} \right\}. \tag{2}$$

The exponential map $\exp : \mathfrak{so}(2) \to SO(2)$ converts an algebra element into a valid rotation matrix:

$$\exp\left( \begin{bmatrix} 0 & -\theta \\ \theta & 0 \end{bmatrix} \right) = \begin{bmatrix} \cos\theta & -\sin\theta \\ \sin\theta & \cos\theta \end{bmatrix}. \tag{3}$$

**SO(2) head for bounded rotation prediction.**    For each frame $X_t$, a lightweight SO(2) head predicts a rotation matrix $R_t \in \text{SO(2)}$. The prediction is parameterized in the Lie algebra using the axis-angle representation $\omega_t \in \mathbb{R}^2$. To avoid degenerate solutions, the rotation magnitude is constrained within a bounded range:

$$R_t = \exp\big( \tanh(\omega_t) \big), \qquad \|\omega_t\| \leq \theta_{\max}, \tag{4}$$

where $\exp(\cdot)$ is the exponential map from $\mathfrak{so}(2)$ to SO(2), and $\theta_{\max}$ specifies the maximum rotation angle. In practice, we use SO(2) rotations by restricting $\omega_t$ to the $z$-axis, which efficiently captures in-plane camera motion.

Compared with directly regressing angles, this design is numerically stable, differentiable, and seamlessly integrates with Lie-theoretic bias construction, which will be detailed in Sec. 3.3.

### 3.3 ROTATION-BOUNDED LIE RELATIVE BIAS

Building on the predicted rotations from the SO(2) head, we construct a *rotation-bounded Lie relative bias* that explicitly injects geometry-consistent information into the attention mechanism.

**Coordinate embedding.** We embed spatial locations into a normalized 3D coordinate basis:

$$P = \{p_i\}_{i=1}^{N}, \quad p_i \in \mathbb{R}^3, \ \|p_i\| = 1, \tag{5}$$

where $p_i$ denotes the position of the $i$-th patch token. For frame $t$, the rotated coordinates are obtained as

$$\tilde{p}_{t,i} = R_t p_i, \tag{6}$$

with $R_t$ provided by the SO(2) head in Sec. 3.2.

**Relative spatial bias.** The spatial bias between token $i$ in frame $t$ and token $j$ in frame $s$ is defined as the inner product of their rotated coordinates:

$$B_{\text{space}}[(t,i),(s,j)] = \langle \tilde{p}_{t,i}, \tilde{p}_{s,j} \rangle, \tag{7}$$

which measures geometry-consistent similarity under the predicted rotations.

**Bias injection.** The computed spatial bias is added to the attention logits:

$$\text{Logits} = \frac{QK^\top}{\sqrt{d}} + B_{\text{space}}. \tag{8}$$

This enables the self-attention to explicitly account for frame-wise poses, guiding the network to aggregate features along geometrically aligned correspondences.

### 3.4 DIFFERENTIAL GROUP DISPLACEMENT

In addition to spatial alignment, temporal consistency is crucial for video restoration. To explicitly capture relative motion between adjacent frames, we introduce a *differential group displacement* based on Lie algebra differences.

**Lie algebra difference.** Given two consecutive frame rotations $R_{t-1}, R_t \in$ SO(2), their relative motion can be represented as

$$\Delta R_t = R_{t-1}^\top R_t. \tag{9}$$

We project $\Delta R_t$ onto the Lie algebra $\mathfrak{so}(2)$ via the logarithm map:

$$v_t = \left\| \log(\Delta R_t) \right\|, \tag{10}$$

where $\log(\cdot)$ denotes the matrix logarithm and $\|\cdot\|$ is the norm measuring the angular displacement. The sequence $\{v_t\}_{t=2}^{T}$ can be interpreted as the Lie-velocity of the video.

**Pairwise angular difference.** For general frame pairs $(t,s)$, the relative angular difference is

$$\theta_{t-1,t} = \left\| \log(R_t^\top R_s) \right\|, \tag{11}$$

which reflects the rotation magnitude required to align frame $t-1$ with frame $t$.

**Temporal bias construction.** We convert the angular difference into a temporal bias that penalizes pairs with large pose discrepancy:

$$B_{\text{time}}[t-1,t] = -\frac{\theta_{t-1,t}}{\kappa}, \tag{12}$$

where $\kappa$ is a scaling constant. This bias is broadcast to the token level and added to the attention logits together with the spatial bias.

**Velocity regularization.** To further stabilize training, we introduce a regularization term on the Lie-velocity sequence:

$$\mathcal{R}_v = (1 - \beta) \cdot \text{mean}(v_t) + \beta \cdot \text{mean}(|v_t - v_{t-1}|), \tag{13}$$

where $\beta$ controls the balance between the overall motion magnitude and smoothness. This regularizer encourages moderate and smooth inter-frame rotations, preventing unstable predictions.

### 3.5 SPATIOTEMPORAL LIE-GROUP DIFFERENTIAL MECHANISM

The final step is to unify spatial and temporal biases into a single mechanism that governs feature aggregation. We refer to this as the *spatiotemporal Lie-group differential mechanism*, which ensures that both geometric alignment and temporal consistency are simultaneously enforced within the attention operation.

**Unified bias.** Given the spatial bias $B_{\text{space}}$ (Section 3.3) and the temporal bias $B_{\text{time}}$ (Section 3.4), the total bias added to the attention logits is

$$B_{\text{total}} = (B_{\text{space}} + \alpha B_{\text{time}}) \odot D \odot M, \tag{14}$$

where $\alpha$ is a scalar balancing temporal bias, $D$ is a temporal decay matrix, and $M$ is a banded mask.

**Temporal decay.** The decay matrix $D \in \mathbb{R}^{T \times T}$ is defined as

$$D[t, s] = \exp\left( - \frac{|t - s|}{\tau} \right), \tag{15}$$

where $\tau$ controls the decay rate. This weighting emphasizes short-range interactions that are more reliable under rain distortions, while gradually suppressing long-range connections.

**Banded attention mask.** The mask $M \in \{0, 1\}^{T \times T}$ restricts each frame to attend only to a local temporal neighborhood:

$$M[t, s] = \mathbb{1}(|t - s| \leq \delta), \tag{16}$$

where $\delta$ specifies the temporal bandwidth. This prevents unstable correspondences between frames that are too far apart.

**Attention with spatiotemporal Lie.** The attention logits for all tokens are finally expressed as

$$\text{Logits} = \frac{QK^\top}{\sqrt{d}} + B_{\text{total}}, \tag{17}$$

with $B_{\text{total}}$ constructed as in Eq. 14. This operation ensures that spatial alignment (via rotated coordinates) and temporal regularization (via angular differences and decay) are tightly coupled inside the same attention layer.

**Loss Function** DeLiVR is trained end-to-end with a hybrid loss that balances reconstruction fidelity and geometric regularization. The reconstruction term $\mathcal{L}_{\text{rec}}$ (L1 loss) ensures pixel-level accuracy between the restored and ground-truth frames. To stabilize pose prediction, we add a rotation magnitude regularizer $\mathcal{R}_\theta$, which constrains the predicted SO(2) rotations, and a Lie-velocity regularizer $\mathcal{R}_v$, which enforces smooth temporal evolution. The final objective is

$$\mathcal{L} = \mathcal{L}_{\text{rec}} + \lambda_\theta \mathcal{R}_\theta + \lambda_v \mathcal{R}_v,$$

where $\lambda_\theta$ and $\lambda_v$ control the trade-off between fidelity and geometric consistency. Based on a grid search for optimal performance, we set the hyperparameter weights to $\lambda_\theta = 0.02$ and $\lambda_v = 0.02$ in our experiments.

## 4 EXPERIMENTS

### 4.1 EXPERIMENTAL SETUP

**Datasets** We conduct a comprehensive evaluation on four benchmark datasets. For synthetic data, we use three widely-adopted benchmarks: NTURain (Chen et al., 2018), Rain-Syn-Light (Liu et al., 2018), and Rain-Syn-Complex (Liu et al., 2018). For real-world evaluation, we use the recently proposed WeatherBench (Guan et al., 2025) benchmark.

**Evaluation Metrics**    We use Peak Signal-to-Noise Ratio, Structural Similarity Index, and Learned Perceptual Image Patch Similarity as the main metrics. Higher PSNR/SSIM and lower LPIPS indicate better video restoration quality. For downstream evaluation, we adopt mean Average Precision for object detection and mean Intersection over Union for semantic segmentation.

**Implementation Details**    Our DeLiVR model is implemented using the PyTorch framework and trained on 8 NVIDIA 3090 GPUs. We use the AdamW optimizer with an initial learning rate of $2 \times 10^{-4}$. The learning rate is decayed over 5000 epochs using a cosine annealing schedule. We set the batch size to 64. For each input, we randomly sample a window of $T = 5$ consecutive frames from the video sequences. Our model is trained with an L1 reconstruction loss, aiming to restore the clean center frame from the rainy input sequence.

## 4.2    COMPARISON WITH STATE-OF-THE-ART METHODS

To comprehensively evaluate the performance of our proposed DeLiVR model, we conduct a comparison with a range of state-of-the-art (SOTA) video deraining algorithms. The selected baselines cover diverse and prominent technical, including classic CNN-based methods (MFGAN (Yang et al., 2021), S2VD (Yue et al., 2021)), Transformer-based models (ESTINe (Zhang et al., 2022), ViWS-Net (Yang et al., 2023)), and other recent architectures (ViMP-Net (Wu et al., 2023), rainmanba (Wu et al., 2024), Turtle (Ghasemabadi et al.), VDMamba (Sun et al., 2025)).

### 4.2.1    QUANTITATIVE ANALYSIS

As shown in Table 1, **DeLiVR** consistently achieves the best overall performance across synthetic and real rainy benchmarks. On NTURain, our method ranks second only to VDMamba (CVPR 2025). More importantly, when evaluated on the real-world WeatherBench dataset, VDMamba suffers a noticeable performance drop (PSNR 23.91, SSIM 0.773), while DeLiVR establishes new state-of-the-art results (PSNR 26.56, SSIM 0.781). This contrast highlights that our Lie-group differential bias design not only provides competitive results on synthetic benchmarks but also yields superior generalization to challenging real rainy scenarios, demonstrating the practical robustness of our approach.

Table 1: Quantitative comparison on synthetic benchmarks and the real-world WeatherBench dataset. The WeatherBench benchmark is included to specifically evaluate the model's generalization ability to authentic, real-world adverse weather conditions. For PSNR/SSIM, higher is better (↑). For LPIPS, lower is better (↓). Best results are in **bold**, and second-best are underlined.

| Method | NTURain | | | Syn-Light | | | Syn-Complex | | | WeatherBench(real-world) | | |
|---|---|---|---|---|---|---|---|---|---|---|---|---|
| | PSNR↑ | SSIM↑ | LPIPS↓ | PSNR↑ | SSIM↑ | LPIPS↓ | PSNR↑ | SSIM↑ | LPIPS↓ | PSNR↑ | SSIM↑ | LPIPS↓ |
| S2VD (CVPR 21) | 32.46 | 0.953 | 0.068 | 25.57 | 0.833 | 0.248 | 16.56 | 0.524 | 0.462 | 26.51 | **0.827** | 0.340 |
| MFGAN (TPAMI 22) | 33.69 | 0.950 | 0.152 | 22.55 | 0.839 | 0.106 | 21.05 | **0.763** | **0.184** | 20.61 | 0.707 | 0.365 |
| ESTINe (TPAMI 23) | 33.93 | 0.950 | 0.035 | 27.61 | 0.879 | 0.169 | 20.16 | 0.653 | 0.302 | 25.22 | 0.795 | **0.334** |
| ViMP-Net (MM 23) | 24.86 | 0.804 | 0.259 | 22.32 | 0.759 | 0.302 | 16.42 | 0.490 | 0.465 | 24.04 | 0.716 | 0.433 |
| ViWS-Net (ICCV 23) | 33.65 | 0.949 | 0.039 | 27.70 | 0.860 | 0.157 | 19.13 | 0.576 | 0.408 | 23.50 | 0.689 | 0.381 |
| rainmanba (MM 24) | 29.48 | 0.876 | 0.194 | 22.71 | 0.749 | 0.295 | 15.59 | 0.474 | 0.470 | 25.38 | 0.763 | 0.391 |
| Turtle (NeurIPS 24) | 23.15 | 0.717 | 0.286 | 21.16 | 0.652 | 0.311 | 14.12 | 0.413 | 0.511 | 22.48 | 0.462 | 0.501 |
| VDMamba (CVPR 25) | **36.29** | **0.973** | **0.010** | 28.76 | 0.896 | 0.157 | 17.83 | 0.582 | 0.376 | 23.91 | 0.773 | 0.344 |
| **DeLiVR (Ours)** | 34.06 | 0.952 | 0.039 | **30.53** | **0.908** | **0.088** | **24.68** | 0.733 | 0.227 | **26.56** | 0.781 | 0.358 |

### 4.2.2    QUALITATIVE ANALYSIS

Figure 3 illustrates the visual comparisons on both synthetic and real-world rainy scenes. Compared to other methods, our DeLiVR is capable of generating sharper images with richer details. For instance, methods like MFGAN and S2VD are prone to producing artifacts or over-smoothed backgrounds. While ESTINe and ViWS-Net are strong performers, they still exhibit residual rain streaks and temporal flickering when dealing with dense rain or fast-moving objects. In contrast, by leveraging geometry-consistent alignment, DeLiVR not only removes rain streaks more thoroughly but also better preserves object edges and fine textures. It also demonstrates stronger temporal consistency in video sequences, without noticeable flickering or artifacts.

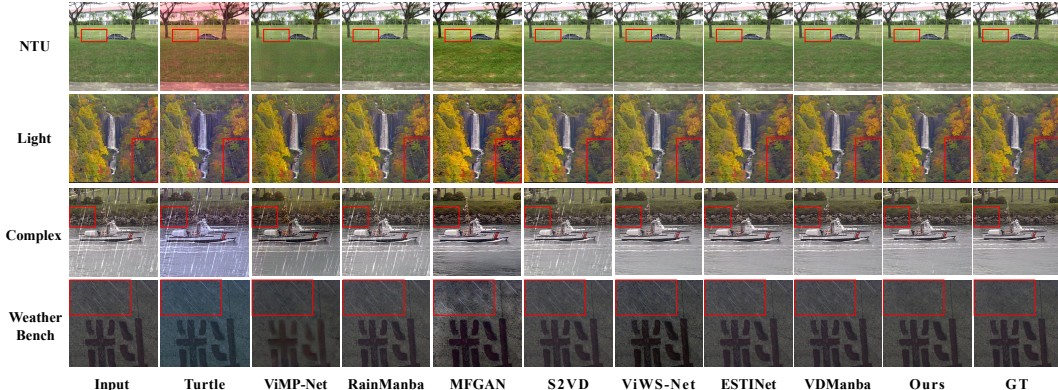

Figure 3: Qualitative comparison with state-of-the-art methods on four benchmarks. From top to bottom, the rows show results on NTU, Rain-Syn-Light, Rain-Syn-Complex, and the real-world WeatherBench dataset. Compared to other methods, our model more effectively removes severe rain streaks and color casts, while better preserving fine background textures and natural colors.

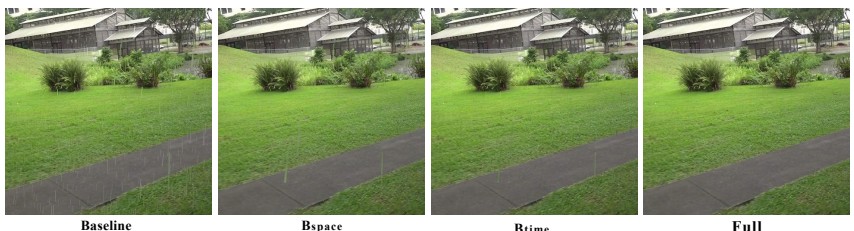

Figure 4: Visual ablation of different bias components on the NTURain dataset. From left to right: baseline without Lie bias, model with spatial bias only ($B_{\text{space}}$), model with temporal bias only ($B_{\text{time}}$), and the full **DeLiVR** with both spatial and temporal Lie-group differential biases.

## 4.3 ABLATION STUDY

To assess the contribution of each component in **DeLiVR**, we conducted an ablation study (Table 2). Starting from a plain spatiotemporal Transformer baseline, adding the spatial bias (**Space**) via rotation-bounded Lie relative bias improved PSNR by 1.37 dB, confirming the importance of explicit geometric alignment. Incorporating the temporal bias (**Time**) through differential group displacement further enhanced performance by modeling inter-frame motion. Finally, adding temporal decay and a banded attention mask (**D&M**) yielded the best results.

Table 2: Ablation study of different components of DeLiVR on the NTURain dataset. For FVD, lower is better ($\downarrow$).

| Model | Baseline | $B_{\text{space}}$ | $B_{\text{time}}$ | D&M | PSNR↑ | SSIM↑ | FVD↓ |
|-------|----------|--------------------|--------------------|-----|-------|-------|------|
| (a) | ✓ | | | | 29.21 | 0.868 | 47.25 |
| (b) | ✓ | ✓ | | | 32.58 | 0.927 | 31.6 |
| (c) | ✓ | ✓ | ✓ | | 33.14 | 0.935 | 22.5 |
| (d) | ✓ | ✓ | ✓ | ✓ | **34.06** | **0.952** | **18.5** |

**Rotation–Perturbation Study.** To directly validate that the proposed Lie-group differential bias improves cross-frame alignment, we conduct a controlled experiment where small in-plane rotations are injected into several frames of the input sequence. This perturbation produces synthetic misalignment while keeping scene content unchanged. As shown in Fig. 5, the rotation-aware model exhibits clearer temporal stability. The attention maps further reveal that the rotation-enhanced model captures richer directional dependencies, displaying higher entropy and more concentrated

high-valued regions. The attention-difference visualization highlights that the Lie-group bias preferentially strengthens correspondences around motion-sensitive structures, thereby reducing flicker and ghosting. These results provide explicit interpretability evidence that our method addresses cross-frame misalignment beyond improving overall metrics.

**Controlled Comparison with Optical Flow Biases.** To provide a comparison between our Lie-group rotation module and optical-flow-based alternatives, we conducted a strictly controlled experiment where all variables except the bias representation were held constant. Specifically, we utilized a 12-layer Transformer backbone and maintained an identical training protocol. For the optical-flow baseline, RAFT flow fields were processed through a small learnable MLP to predict bias maps, mirroring the structure of our rotation module. This design ensures the *only* differing factor is the choice of bias representation. Evaluations on the NTU-Rain dataset show our method improves PSNR by +2.43 dB, demonstrating that modeling rain-streak orientation on the $SO(2)$ manifold provides a stronger and more stable geometric inductive bias compared with unconstrained optical-flow fields. Visualizations in in Appendix (Fig. 9) and real-world results in Appendix (Fig. 10) further confirm that our proposed prior excels on benchmarks and transfers robustly to naturally occurring rain conditions.

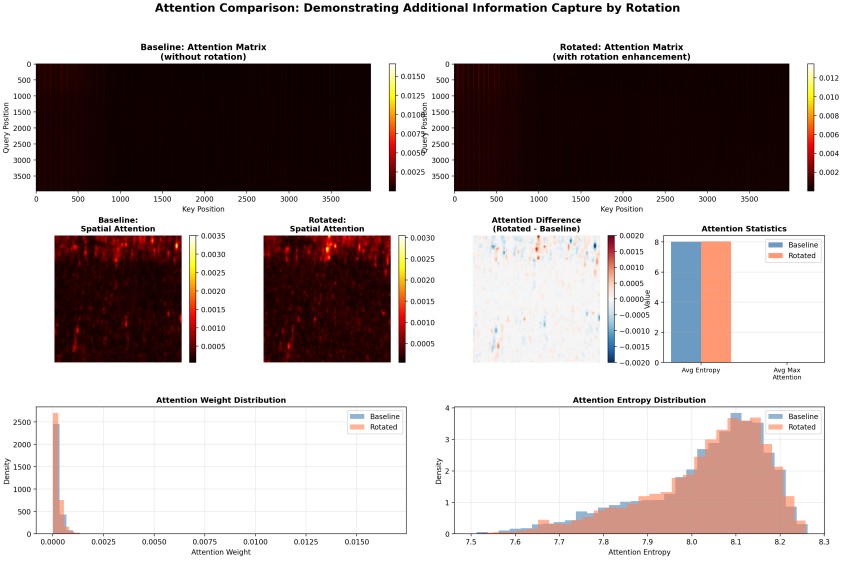

Figure 5: **Attention comparison between the baseline and the rotation-enhanced model.** The top row shows full attention matrices for both models under the same input sequence. The second row visualizes spatial attention maps extracted from representative query positions. The right column reports aggregated attention statistics, including average entropy and maximum attention values. The bottom row presents the distribution of attention weights and attention entropy across all heads and layers.

## 4.4 DISCUSSION

**Impact on Downstream Tasks** To assess the practical value of our deraining method, we tested its impact on downstream tasks like object detection and semantic segmentation. As visually demonstrated in Figure 6, adverse weather conditions severely degrade the performance of these high-level vision models. The top row shows that the object detector fails on the rainy input, but after processing with our method, it can accurately localize the targets. Similarly, for semantic segmentation (bottom row), our method helps the model produce a much cleaner and more precise mask compared to the corrupted result from the original image. This intuitively proves that our method not only enhances visual quality but also acts as a crucial pre-processing step to improve the reliability of downstream vision systems in real-world scenarios.

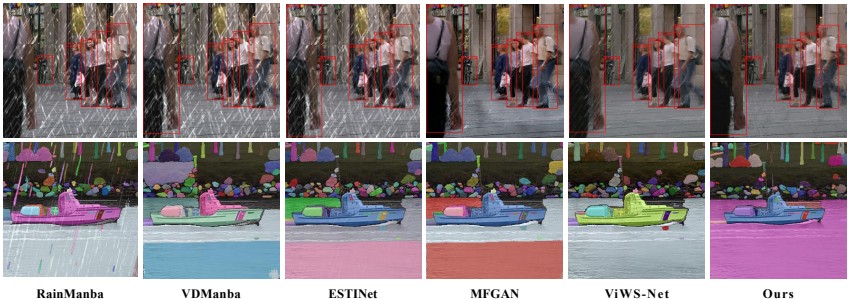

Figure 6: Visual comparison of downstream task performance on the Rain-Syn-Complex dataset.

**Efficiency Analysis.** In addition to restoration accuracy, efficiency plays a crucial role in the practical deployment of video deraining systems. Table 3 reports the model size (in terms of parameters) and the average inference time per frame. DeLiVR demonstrates clear advantages over prior methods: it uses significantly fewer parameters and achieves much faster inference, while still delivering competitive or superior restoration quality. This confirms that introducing Lie-group differential biases is not only theoretically meaningful and robust, but also lightweight and computationally efficient, making DeLiVR more suitable for real-world applications.

Table 3: Efficiency comparison on the NTURain dataset. Params denote model parameters (in millions). Time denotes average inference time (ms per frame).

| Metric | rainmanba | ViMP-Net | Turtle | ViWS-Net | VDMamba | ESTINe | MFGAN | S2VD | **DeLiVR (Ours)** |
|---|---|---|---|---|---|---|---|---|---|
| Params (M)↓ | 38.73 | 36.63 | 58.62 | 57.82 | 12.70 | 29.90 | 29.47 | **0.53** | 2.64 |
| Time (ms)↓ | 145.55 | 130.77 | 270.54 | 252.64 | 32.04 | 175.61 | 162.81 | **26.78** | 82.52 |

**Comparison with Optical-Flow-Based Guidance.** We evaluate two representative flow-guided recurrent video restoration models on the WeatherBench dataset: *Frame-Consistent Recurrent Video Deraining (Yang et al., 2019)* (PSNR 23.33 / SSIM 0.691 / LPIPS 0.455) and *High-Resolution Optical Flow and Frame-Recurrent Network (Fang & Zhan, 2022)* (PSNR 24.10 / SSIM 0.716 / LPIPS 0.402). Under the same evaluation setting on an A100 GPU, our rotation-prediction-based model consistently surpasses both flow-based baselines.

**Necessity of Lie Groups.** Our design leverages Lie-group theory to ensure valid and continuous rotations, provide a tangent space for stable optimization, and enable principled definitions of temporal displacement. Although similar effects could be superficially achieved by direct matrix parameterization, the Lie-group formulation offers theoretical rigor and extensibility. A detailed analysis is provided in Appendix A.1.

**Limitations.** Despite its effectiveness, DeLiVR has certain limitations. First, the reliance on rotation-centric modeling may not fully capture more complex non-rigid rain dynamics or camera motions beyond in-plane rotations. Second, incorporating Lie-group biases introduces additional computational overhead compared to purely implicit alignment schemes. Addressing these challenges and extending the framework to richer transformation groups (e.g., $SE(2)/SE(3)$) are promising directions for future work.

## 5 CONCLUSION

We proposed DeLiVR, a video deraining framework that injects Lie-group differential biases into attention to achieve geometry-consistent alignment and motion-aware temporal modeling. Our method is theoretically grounded through the use of Lie algebra for stable optimization and principled temporal displacement, and empirically validated by extensive experiments. Experimental results demonstrate that DeLiVR not only achieves state-of-the-art performance on real rainy datasets but also enhances downstream tasks such as object detection and semantic segmentation. These findings highlight both the superior robustness of DeLiVR compared with existing methods and the practical value of integrating geometric theory into attention for reliable video restoration.

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

## A  APPENDIX

### A.1  THEORETICAL ANALYSIS OF LIE GROUP FORMULATION

Although one may use rotation matrices to approximate rain streak orientation, here we provide a theoretical analysis showing why a Lie group formulation is essential for our task.

**(i) Guaranteed validity of rotations.**  A valid planar rotation must lie in the special orthogonal group:
$$SO(2) = \{R \in \mathbb{R}^{2\times2} \mid R^\top R = I, \det(R) = 1\}. \tag{18}$$
If $R$ is directly parameterized as an unconstrained matrix, optimization may yield invalid results (e.g., $\det(R) \neq 1$ or $R^\top R \neq I$). Enforcing orthogonality requires either costly projection steps or implicit normalization, which are non-trivial in gradient-based learning. By contrast, parameterizing in the Lie algebra $\mathfrak{so}(2)$ with exponential mapping
$$R = \exp(\omega), \qquad \omega \in \mathfrak{so}(2), \tag{19}$$
guarantees $R \in SO(2)$ by construction. Thus, every predicted transformation remains a mathematically valid rotation.

**(ii) Tangent space for stable optimization.**  The Lie algebra $\mathfrak{so}(2)$ is the tangent space at the identity, defined as
$$\mathfrak{so}(2) = \left\{ \begin{bmatrix} 0 & -\theta \\ \theta & 0 \end{bmatrix} : \theta \in \mathbb{R} \right\}. \tag{20}$$
This provides a *linear* space for optimization, allowing unconstrained updates $\omega \in \mathbb{R}$ while ensuring stability. Optimization in this Euclidean tangent space avoids the degeneracy and instability of directly manipulating matrix entries subject to nonlinear constraints.

**(iii) Principled definition of temporal displacement.** For temporal modeling, we require a consistent notion of relative rotation between frames. Given two orientations $R_t, R_s \in SO(2)$, their relative transformation is

$$\Delta R_{t,s} = R_t^\top R_s \in SO(2). \tag{21}$$

The logarithm map $\log : SO(2) \to \mathfrak{so}(2)$ then yields

$$v_{t,s} = \|\log(\Delta R_{t,s})\|, \tag{22}$$

which quantifies the angular displacement as a scalar in the Lie algebra. This is a principled, geometrically meaningful notion of motion, while subtracting raw matrices $R_t - R_s$ provides no intrinsic interpretation of displacement.

In summary, while plain rotation matrices can approximate orientation, the Lie group perspective (i) ensures validity of rotations by construction, (ii) offers a tangent space for stable gradient-based optimization, and (iii) enables principled temporal displacement via group differences and logarithm mapping. These theoretical properties justify our design choice and highlight why Lie groups are essential in modeling spatiotemporal alignment for video deraining.

## A.2 CORE PSEUDOCODE OF DELIVR

Algorithm 1 summarizes the overall workflow of our proposed DeLiVR framework. The process begins with patch embedding to obtain tokenized representations, followed by frame-wise rotation prediction through the SO(2) head. The predicted rotations are then used to construct spatial and temporal biases, which are fused into a total bias and injected into the attention mechanism. Finally, the updated token states are decoded to reconstruct the clean center frame, with the entire model trained under a joint reconstruction and regularization loss.

---

**Algorithm 1** Core pseudocode of DeLiVR

---

1: **Input:** Rainy video window $\mathbf{X} = \{X_t\}_{t=1}^T$
2: **Output:** Restored center frame $\widehat{Y}_c$
3: Patchify all frames and obtain initial embeddings $\mathbf{H}^{(0)}$.
4: For each frame $X_t$, use SO(2) head to predict rotation $R_t$.
5: Construct biases:
      1. Rotated coordinates: $\tilde{p}_{t,i} = R_t p_i$.
      2. Spatial bias: $B_{\text{space}}[(t,i),(s,j)] = \langle \tilde{p}_{t,i}, \tilde{p}_{s,j} \rangle$.
      3. Temporal bias: $B_{\text{time}}[t,s] = -\|\log(R_t^\top R_s)\|/\kappa$.
      4. Fuse: $B_{\text{total}} = (B_{\text{space}} + \alpha B_{\text{time}}) \odot D \odot M$.
6: Update tokens for $l = 1, \dots, L$:
   $\mathbf{H}^{(l+1)} = \mathbf{H}^{(l)} + \text{Attention}(\mathbf{H}^{(l)}; B_{\text{total}})$.
7: Decode center tokens to obtain $\widehat{Y}_c$.
8: Train with $\mathcal{L} = \mathcal{L}_{\text{rec}} + \lambda_\theta \mathcal{R}_\theta + \lambda_v \mathcal{R}_v$.

---

## A.3 GENERALIZATION TO DEBLURRING AND DEHAZING

Although our method is introduced primarily for video deraining, the core idea—injecting a Lie-group differential bias to stabilize cross-frame alignment—is degradation-agnostic. To further verify this, we additionally evaluate the model on two representative degradation types: **deblurring** (trained on the GoPro dataset (Nah et al., 2017)) and **dehazing** (trained on the RESIDE dataset Li et al. (2019)). As shown in Fig. 7, our method consistently improves *geometric stability*, reduces *temporal flickering*, and preserves *fine structures* across all three tasks (deraining, deblurring, dehazing). These results demonstrate that the proposed geometric prior captures fundamental motion characteristics rather than degradation-specific cues, thereby enabling natural transfer to a broader range of video restoration scenarios.

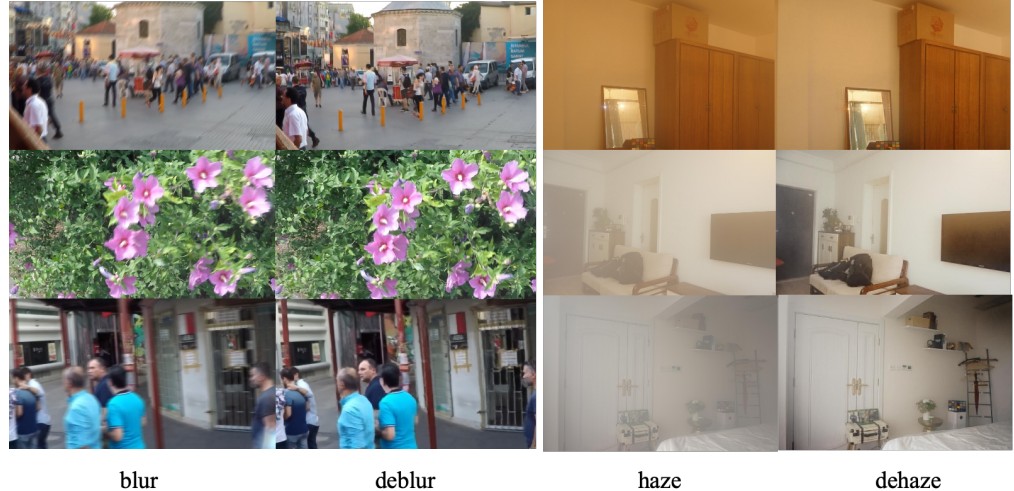

| blur | deblur | haze | dehaze |

Figure 7: **Generalization to other degradations.** Our method consistently produces cleaner structures and fewer temporal flickers.

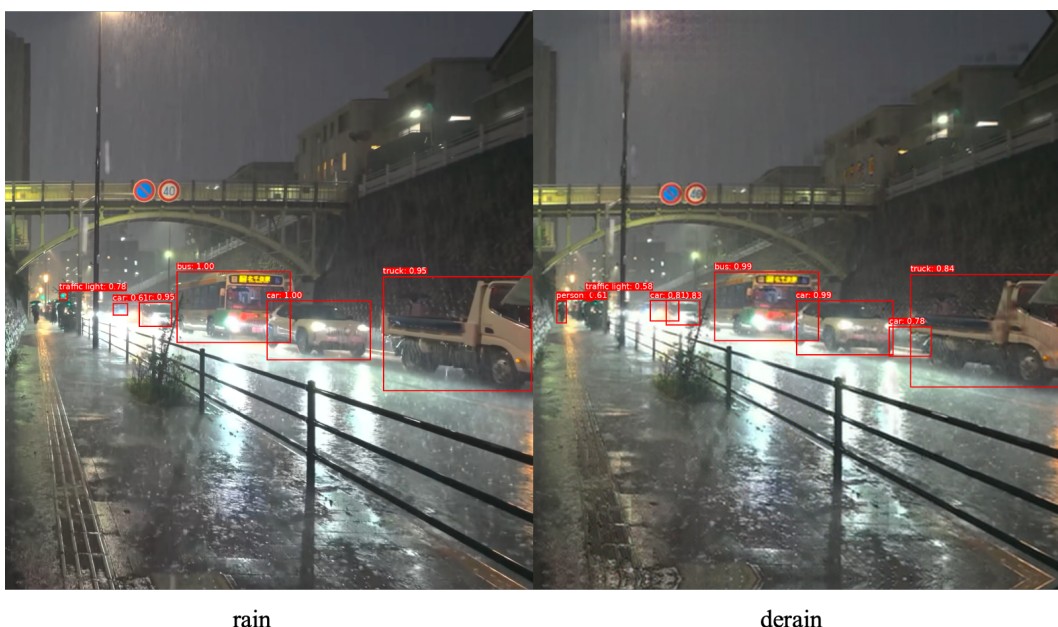

| rain | derain |

Figure 8: **Real-world downstream evaluation on object detection.** We apply an off-the-shelf detector to a real captured rainy frame (left) and to the derained result produced by our method (right).

## A.4 REAL-WORLD DOWNSTREAM EVALUATION

To further validate the practical utility of our method, we test an off-the-shelf object detector on a real captured rainy image. As shown in Fig. 8, deraining enables the detector to identify an additional pedestrian missed in the rainy input, demonstrating improved reliability for downstream perception.

## A.5 LARGE VISUALIZATION OF FIGURE 3

To provide a clearer view, we include the enlarged version of Figure 3 in Appendix.

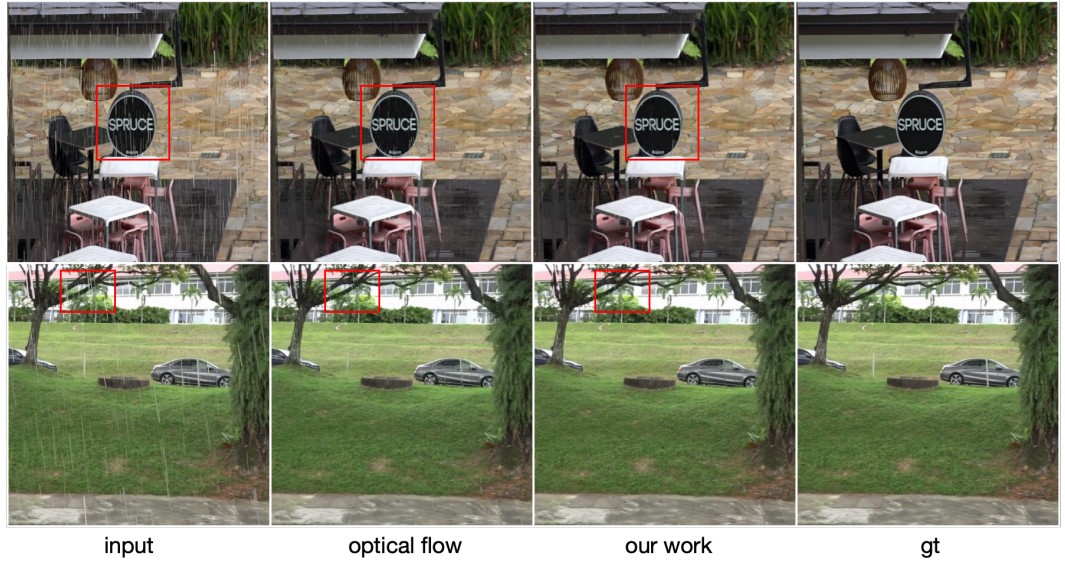

Figure 9: **Visual comparison on the NTU-Rain.** From left to right: rainy input, flow-based bias baseline, our Lie Rotation model, and ground-truth.

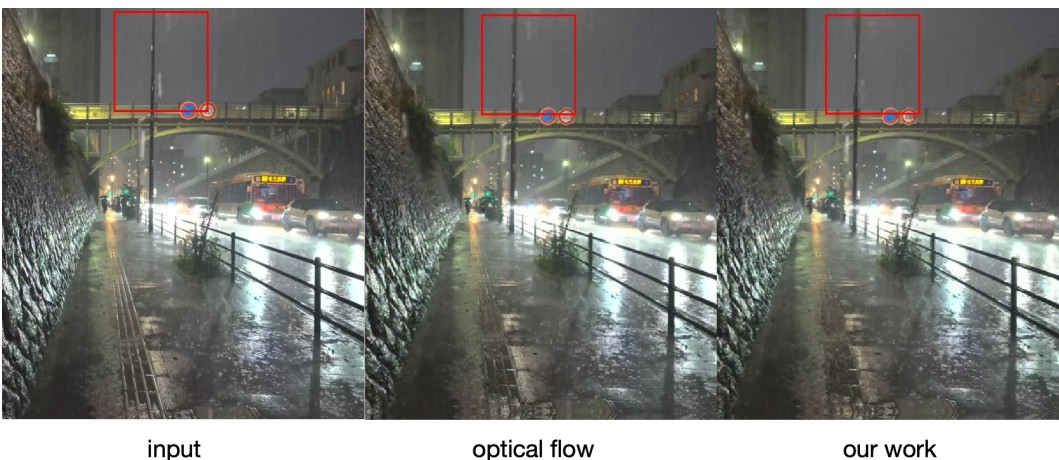

Figure 10: **Real-world rain removal.** From left to right: the original rainy input, the flow-based bias baseline, and our Lie Rotation model.

### A.6 USE OF LARGE LANGUAGE MODELS

During the preparation of this paper, large language models (LLMs) were employed for *textual refinement*, including grammar checking, clarity improvement, and stylistic polishing of the manuscript.

### ETHICS STATEMENT

This work focuses on video deraining, a low-level vision task, and does not involve human subjects or sensitive personal data. All datasets used in our experiments are either publicly available synthetic datasets or real-world benchmarks that do not contain personally identifiable information. Our method does not introduce risks of privacy leakage or misuse beyond the scope of general image

and video restoration. We adhere to the ICLR Code of Ethics, and we believe that our contributions advance fundamental research in computer vision without posing foreseeable ethical concerns.

### REPRODUCIBILITY STATEMENT

We have made significant efforts to ensure the reproducibility of our work. The paper provides detailed descriptions of the proposed architecture, bias formulation, and loss functions in Sections 3, with theoretical derivations further explained in the appendix. Implementation details, training setups, and hyperparameters are described in the experimental section and supplementary materials. Pseudocode for the core algorithm is included in the appendix (Algorithm 1), and all datasets used (NTURain, Syn-Light, Syn-Complex, WeatherBench) are publicly available.

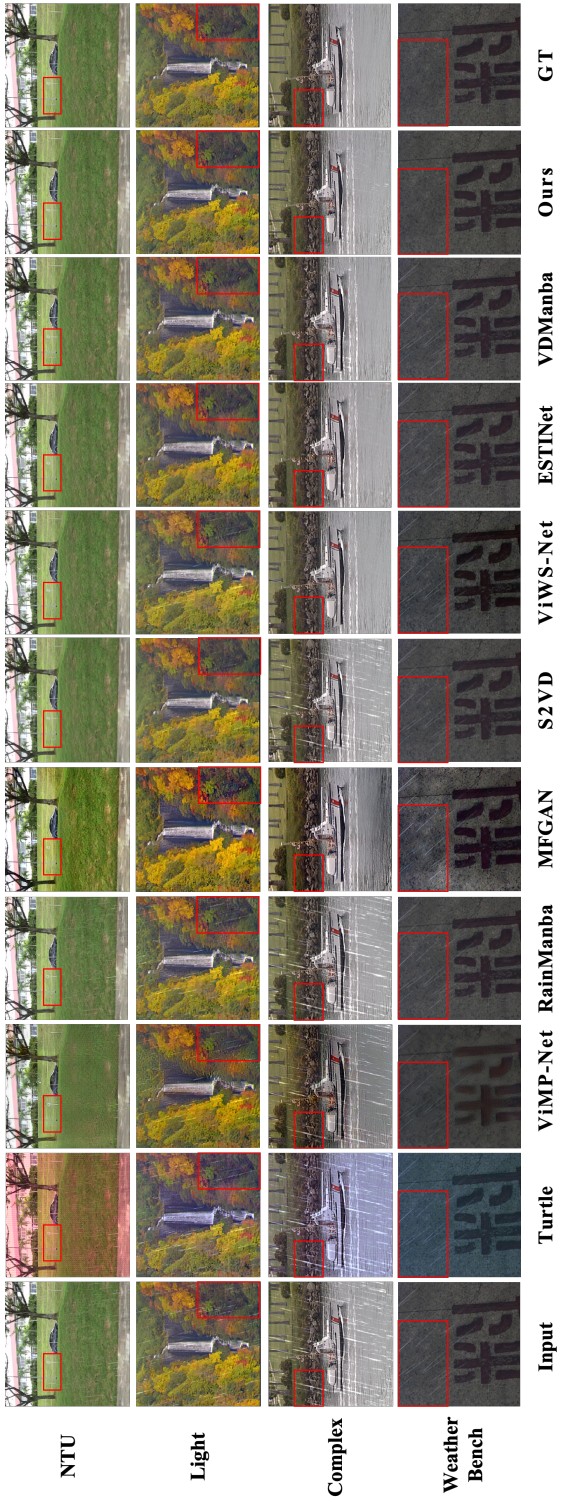

Figure 11: Enlarged visualization corresponding to Figure 3.

