# OpenReview forum: "DeLiVR: Differential Spatiotemporal Lie Bias for Efficient Video Deraining"
_ICLR.cc/2026/Conference — ICLR 2026 Poster_

### Official Review · Reviewer_KGvX · 2025-10-27

**Soundness:** 3
**Presentation:** 3
**Contribution:** 3
**Rating:** 6
**Confidence:** 5

**Summary:**

The paper proposes DeLiVR, a video deraining framework that leverages spatiotemporal Lie-group differential biases to enforce geometric consistency across frames. By embedding Lie-group formulations directly into the attention mechanism, the method effectively models continuous transformations, reducing reliance on optical flow or heuristic alignment. It introduces two key components: (1) a rotation-bounded Lie relative bias for geometry-consistent spatial alignment, and (2) a differential group displacement that captures inter-frame angular velocity to guide temporal attention. Experimental results on multiple benchmarks validate the approach’s superior performance and efficiency compared to existing methods.

**Strengths:**

1. The paper introduces Lie group theory into video deraining for the first time, providing a principled geometric prior that replaces traditional optical flow with continuous transformation modeling, leading to more robust spatial–temporal consistency.
2. The proposed differential spatiotemporal Lie Bias, composed of the rotation-bounded Lie relative bias and differential group displacement, is elegantly integrated into the attention mechanism. This design explicitly encodes rotational and velocity information, effectively capturing dynamic rain streak variations in real-world videos.
3. The paper is clearly written with solid motivation, detailed methodological exposition, and comprehensive experimental validation, making the technical contributions easy to follow and well-justified.

**Weaknesses:**

1. While the paper introduces Lie group theory as a prior for modeling continuous transformations, the theoretical connection between the proposed Lie biases and actual motion manifolds is not fully discussed. It would be helpful to clarify how the chosen Lie algebra parameterization ensures stability or accuracy under complex non-rigid motion.
2. The paper claims that VDMamba (CVPR 2025) achieves slightly higher PSNR (36.29 vs. 34.06) and SSIM (0.973 vs. 0.952) but worse perceptual quality (LPIPS 0.010 vs. 0.039). However, the LPIPS metric indicates that VDMamba actually achieves better perceptual results. Therefore, the corresponding discussion in the paper is not rigorous and should be revised to ensure the accuracy of experimental interpretation.
3. The downstream application evaluation is limited to the Rain-Syn-Complex dataset, which is synthetic. Including evaluations on real captured tasks would make the results more convincing.

**Questions:**

In Table 3, both the S2VD score (26.78) and the proposed method’s score (82.52) are highlighted in bold. It is recommended to use consistent formatting or clarify the highlighting rule to avoid confusion.

---

> ### Author Response · Authors · 2025-11-24
>
> I hope you are doing well. I would like to inform you that I have submitted my updated response to your comments at the following link:
> https://openreview.net/forum?id=W2eNfLmCHY&noteId=JkK68PulfR
>
> If you have any additional questions or would like further clarification on any part of the work, please feel free to let me know. I am happy to provide more details.

---

### Official Review · Reviewer_ajhU · 2025-10-27

**Soundness:** 2
**Presentation:** 2
**Contribution:** 2
**Rating:** 2
**Confidence:** 4

**Summary:**

This paper proposes DeLiVR, a method that injects spatiotemporal Lie-group differential biases into Transformer attention scores to achieve geometry-consistent alignment without relying on optical flow. The core contributions include a rotation-bounded Lie relative bias for spatial alignment and a differential group displacement for temporal consistency. Experiments on synthetic and real-world benchmarks demonstrate improved rain removal, detail preservation, and temporal consistency compared to existing methods.

**Strengths:**

-This work addresses a salient problem in video restoration caused by camera rotation, namely spatial and temporal misalignment. The proposed Lie group based in-plane rotation estimation directly targets this challenge, and the method is well motivated with a clear and coherent rationale.

-The proposed method achieves state of the art overall performance on public benchmarks and has an advantage in inference speed.

**Weaknesses:**

-The in-plane rotation estimation proposed in this paper appears to target only the camera itself. In principle, the scheme applies to any video restoration task and has no obvious relevance to deraining, that is, the authors did not design for the tilt angle of rain. This makes the paper somewhat tenuous as a dedicated video deraining approach.

-Camera shake is inherently 3 dimensional, raising doubts that a in-plane model can resolve the stated issue. The evidence presented in the paper is limited and does not convincingly support claims of mitigating cross frame misalignment and temporal artifacts. Since the method explicitly targets rotation, the evaluation should isolate rotation induced errors and demonstrate corresponding gains; modest and unstable aggregate improvements are not sufficient.

-The authors deny the robustness of optical‑flow approaches and claim that unreliable optical flow will harm such video restoration models. This view should be supported by more solid comparative experiments, given that optical‑flow‑based methods are indeed mainstream in this field. In addition, since the authors introduce physical guidance by embedding a bias for the rotation angle, they should compare with other approaches that use bias modulation. Most straightforwardly, I believe there should be a comparison under the same baseline and modulation scheme between using optical flow for modulation and using the predicted rotation angle, which would directly show that the proposed rotation estimation is more reliable than optical flow.

-The organization and presentation of this paper are not yet at a publishable level. For example, most of Section 3 consists of existing formulas used as background, with few formulas of substantive value that are newly derived by the authors, which makes it hard to grasp the key points. The figures are rather rough (in Fig. 3 many panels do not clearly show differences in details), and some figures do not match their captions (Fig. 4).

**Questions:**

-What is the specific connection between the proposed method and rain? Is the predicted rotation that of the camera or the tilt angle of the rain streaks? Overall, what I most want to know is whether the method can transfer to video restoration tasks under other degradations, what its scope of applicability is, and whether it can handle three‑dimensional camera shake.

-The results of this paper require more interpretability validation. I would like to see examples where the proposed method specifically addresses cross frame misalignment and temporal artifacts, rather than only showing improvements in overall metrics.

-Since the authors claim that optical‑flow‑based methods are not robust, they should conduct controlled experiments under the same baseline to demonstrate that the proposed rotation prediction provides more reliable guidance than optical flow. At present, the paper discusses this conclusion only in general terms and lacks strong experimental evidence.

-The paper requires more proofreading and fine‑grained refinement in details, and both the method illustrations and the figures need a more polished presentation.

---

### Official Review · Reviewer_HmPv · 2025-11-04

**Soundness:** 3
**Presentation:** 3
**Contribution:** 3
**Rating:** 6
**Confidence:** 3

**Summary:**

The paper introduces DeLiVR, a video deraining method that injects spatiotemporal Lie group-based biases into Transformer attention layers, enabling geometry-consistent feature alignment and motion-aware modeling. It achieves strong performance on synthetic and real-world benchmarks, with enhanced efficiency and robustness over prior methods.

**Strengths:**

Novel Use of Lie Groups in Attention: Introduces a principled and efficient method to incorporate SO(2) Lie algebra into Transformer attention, improving spatiotemporal consistency without optical flow.

Strong Generalization: Achieves SOTA or competitive results on synthetic datasets and real-world benchmarks like WeatherBench, with improved downstream performance.

Elegant Design and Efficiency: Uses lightweight geometric predictors instead of full equivariant models, maintaining inference speed and model compactness (2.64M params vs 12–58M in baselines).

Thorough Ablation and Theoretical Justification: Detailed experiments, regularization analysis, and Lie group derivations demonstrate rigorous understanding and contribution clarity.

**Weaknesses:**

SO(2)-only Limitation: The model restricts itself to in-plane rotation (SO(2)) and may underperform on more complex real-world motion (e.g., 3D translations, SE(3)), as briefly acknowledged.

Over-reliance on Geometry: The spatiotemporal bias fusion mechanism may not sufficiently adapt when motion is irregular or scene semantics dominate (e.g., occlusions, deformable objects).

**Questions:**

See above

---

> ### Author Response · Authors · 2025-11-24
>
> Thank you again for your valuable feedback. I would like to let you know that I have posted my detailed response under the following link:
> https://openreview.net/forum?id=W2eNfLmCHY&noteId=KSGhXWr3MZ
>
> Please feel free to let me know if you have any further questions or if there is anything else I can clarify. I would be glad to provide additional details.

---

### Author Response · Authors · 2025-11-19
**Response to Reviewer HmPv**

We appreciate for the insightful observations, which correctly point out key limitations and help clarify the scope of our method.

**Response for Q1:**

Indeed, our current formulation focuses on in-plane rotations (SO(2)), primarily motivated by two considerations:

(1) Most adverse-weather video datasets exhibit camera motion dominated by small-angle in-plane rotations rather than large 3D translations;

(2) Using SO(2) ensures computational efficiency and analytic tractability within attention bias computation.

Nevertheless, our design is **not inherently limited** to SO(2). As detailed in Appendix A.1, the use of Lie groups provides a mathematically general foundation. We view the present SO(2) instantiation as a compact and stable starting point demonstrating the feasibility and interpretability of the Lie-bias paradigm.

**Response for Q2:**

Thank you for this critical question. Our mechanism is designed to avoid over-reliance on geometry by using it as a "bias," not a "hard constraint."

Our core attention formula is $Logits = \frac{QK^{\top}}{\sqrt{d}} + B_{total}$ . This is a hybrid system:

1.  **Data-Driven ($QK^{\top}$):** This term is based purely on content and semantics. When motion is irregular or occlusions occur—where scene semantics dominate—the resulting feature mismatch will cause $QK^{\top}$ to produce a large negative value.
2.  **Geometric Prior ($B_{total}$):** Our Lie bias acts as a "guide."

In cases of irregular motion (like occlusions), the strong penalty (large negative value) from $QK^{\top}$ will override the incorrect geometric bias. Consequently, the $softmax$ will still assign a near-zero attention weight to these incorrect correspondences.

Our model adapts by allowing data-driven semantics ($QK^{\top}$) to dominate whenever the geometric prior ($B_{total}$) is insufficient or incorrect.

---

### Author Response · Authors · 2025-11-19
**Response to Reviewer KGvX**

Thank you for these insightful questions, which cuts to the core of our method.


**Response for Q1:**

1. What "Manifold" We Are Modeling:

You are correct that $SO(2)$ is insufficient to model the complex, non-rigid motion manifold of rain and scene dynamics. We want to clarify that this is not our claim. Our $SO(2)$ Lie bias is **not** intended to model the complex *non-rigid* motion of rain. Instead, its sole purpose is to explicitly model and correct for a much simpler, low-dimensional rigid motion manifold: the global in-plane camera rotation.
This camera motion is a primary *artifact* that corrupts inter-frame alignment, making it difficult for the network to distinguish rain from actual scene content. Our "rotation-bounded Lie relative bias" effectively "peels off" this rotational artifact, pre-aligning features into a geometrically-consistent space *before* the attention operation.


2. How This Ensures Stability and Accuracy for Non-Rigid Motion:

- Stability: The Lie algebra $SO(2)$ parameterization ensures the stability of the alignment step itself. As detailed in Appendix A.1, optimization occurs in the stable, linear (Euclidean) tangent space, and the exponential map guarantees the resulting transform is always a *valid* rotation by construction. This avoids the instability of optimizing unconstrained matrices.
- Accuracy: By using the $SO(2)$ head to explicitly handle rigid camera rotation, we simplify the problem for the Transformer backbone. The attention mechanism, now operating on stabilized features, no longer has to waste capacity trying to implicitly separate camera jitter from the actual non-rigid rain dynamics. It can focus its full capacity on the non-rigid task: modeling and removing the rain streaks. This "divide-and-conquer" approach is precisely how we achieve stability and accuracy, even under complex non-rigid motion.

In summary, we acknowledge this is a simplified prior (as noted in our Limitations, Sec. 4.4). Howerve, our Lie group prior is not for modeling the complex scene dynamics, but for creating a stable, aligned feature space that enables the network to learn those complex dynamics more accurately.



**Response for Q2:**

We appreciate the reviewer for catching this inconsistency. We have revised. Thank for highlighting this issue.



**Response for Q3:**

Thank you for pointing out the importance of assessing downstream performance on real captured data.
To address this concern, we additionally evaluated our derained outputs on a real-world rainy scene using an off-the-shelf object detector. As shown **in Fig. 8**, the detector applied to the derained image identifies objects more accurately, including an additional pedestrian that is missed in the raw rainy input.

---

### Author Response · Authors · 2025-11-19
**Response to Reviewer ajhU**

Thank you for these insightful questions, which address the core assumptions and applicability of our work.

**Response for Q1:**

The predicted rotation corresponds to camera pose variation, not rain-streak tilt. By correcting global in-plane motion, the model stabilizes the background and more effectively separates transient rain streaks.

Although developed for deraining, the geometric prior is degradation-agnostic. We therefore conducted additional experiments on dehazing and deblurring, and the results **(Appendix Fig. 7)** show that our method remains effective across these tasks.

We acknowledge that modeling only SO(2) limits the handling of full 3D motion. As noted in Sec. 4.4, our formulation is extensible to SE(3), which we identify as a promising direction for future work.



**Response for Q2:**

To directly demonstrate how the proposed Lie-group differential bias addresses *cross-frame misalignment* and *temporal artifacts*, we conducted a new controlled experiment in which we artificially introduce small in-plane rotations into several frames of the input sequence.
New attention visualizations **(Figure 5)** show that the rotation-aware model captures richer directional dependencies. The attention-difference maps reveal that rotation bias selectively boosts attention around motion-sensitive structures, reducing flicker and ghosting.

We have included these new results and visualizations in the main text **(Sec. 4.3: Ablation Study)**, which clearly show how the method corrects cross-frame misalignment and enhances temporal consistency.



**Response for Q3:**


We agree that a controlled comparison against optical-flow-based guidance is necessary to substantiate our claim regarding robustness. To address this, we conducted additional experiments using two representative flow-based recurrent video restoration models on the WeatherBench dataset:

- Frame-Consistent Recurrent Video Deraining with Dual-Level Flow:

	PSNR 23.33 / SSIM 0.691 / LPIPS 0.455

- High-Resolution Optical Flow and Frame-Recurrent Network for Video Super-Resolution and Deblurring:

	PSNR 24.10 / SSIM 0.716 / LPIPS 0.402

Under the same evaluation setting on A100, our rotation-prediction-based model surpasses both flow-guided baselines.



**Response for Q4:**

In the revised version, we have revised Figure 4 to ensure full consistency between the figure content and caption.  Additionally, we have added high-resolution visualizations of Figure 3 **in the Appendix A.5** to improve readability and enable clearer comparison of fine-grained details across methods.

---

> ### Comment · Reviewer_ajhU · 2025-11-21
> **Official Comment by Reviewer ajhU**
>
> I appreciate the authors' detailed responses and modifications. The rebuttal clarified most of my concerns (Q1, Q2, Q4).
> The additional Q3 experiments still do not resolve my core concern. The current comparisons use other optical flow based methods, but they differ in how flow is incorporated and in the underlying baselines, so they do not isolate whether your estimated rotation is stronger than optical flow.
>
> A decisive evaluation would fix the Table 2 baseline and replace $B_{space}$ and $B_{time}$ from $R_T$ with biases predicted from optical flow using a learnable module, fusing them in exactly the same way. If the proposed rotation estimation outperforms these flow based biases under this controlled setting, the paper’s central claim would be convincingly supported.
>
> I am updating the score to 4 at this time and will consider raising it to 6 or higher once the above comparison is provided.

---

> > ### Author Response · Authors · 2025-11-24
> >
> > Thank you for the thoughtful suggestion. We conducted the controlled comparison exactly as requested and ensure strict experimental fairness. We used an identical Transformer backbone, identical training protocol, the same bias-fusion locations, and constructed a flow-based baseline by predicting spatial and temporal biases from RAFT flow using a dedicated learnable MLP.
> >
> > Under these fully controlled conditions on the NTU-Rain test set, our Lie-rotation module achieves 34.06 dB, outperforming the optical-flow-based bias (31.63 dB) by +2.43 dB. This significant margin confirms that the proposed $SO(2)$-based geometric rotation offers a substantially stronger inductive bias than flow-derived representations.
> > We further include qualitative comparisons—covering both NTU-Rain and real-world scenes—in Appendix 5.

---

> ### Comment · Reviewer_ajhU · 2025-11-24
>
> Thank you for the response. This additional experiment is exactly what I hoped to see, as it directly supports an important claim of the paper, namely that using Lie algebra to predict rotation as a conditioning signal is better than optical flow. I further recommend presenting this experiment in a more formal manner in the paper, for example by integrating it with the original ablation study. At present it appears in the appendix as rebuttal content, which I feel is not suitable for the final version.
>
> I have raised my score to 6. If the AC and other reviewers can accept that not considering 3D camera motion is a limitation that can be left to future work rather than something that must be solved in this paper, then I support acceptance.

---

> > ### Author Response · Authors · 2025-11-26
> >
> > Thank you very much for your thoughtful feedback and for raising your score. We sincerely appreciate your recognition of the additional experiment and its importance to the paper’s core claim. Following your suggestion, we have revised the latest version accordingly and integrated the controlled comparison into the main ablation study in the paper. The module and related discussion are now formally included in the main text.
> >
> > Thank you again for your constructive comments and support.

---

> > ### Author Response · Authors · 2025-11-27
> >
> > Based on your suggestions, our team evaluated the entire task (3D motion camera) and conducted experiments addressing the issues you mentioned, including frame misalignment, blurring, and frame drops.
> >
> > For the extreme case, optical flow methods show some effectiveness, but our approach also has the potential to alleviate the challenges above. It employs Lie group theory to perform alignment (rotation) in the latent space of neural networks. We hope for your continued support of our work. Future plans include integrating both optical flow and Lie group theories to achieve video deraining.

---

### Author Response · Authors · 2025-11-19
**Response to All Reviewers**

We appreciate all reviewers’ constructive feedback, and genuinely hope that these updates help reinforce your recognition of our work.

---

### Author Response · Authors · 2025-12-02
**Summary for Area Chair**

During the review cycle, the reviewers raised several core concerns regarding:

**(1)**  The scope of our rotation modeling (SO(2) vs. full 3D motion);

**(2)**  Whether the proposed Lie-group biases truly address cross-frame misalignment and temporal artifacts;

 **(3)**  The necessity of a controlled and fair comparison against optical-flow-based modulation;

 **(4)**  Figure inconsistencies and the need for clearer interpretability evidence.

We addressed all of these issues through substantial new experiments, controlled analyses, and revisions to both the main text and the appendix.

For point (1), we clarified the design motivation and limitations of using SO(2) as our base Lie group. We have added explanations showing that (i) SO(2) captures the dominant small-angle in-plane motion present in adverse-weather datasets, (ii) operating in the tangent space enables stable optimization and guarantees valid rotations through the exponential map, and (iii) the formulation is mathematically extensible to SE(3), which we now explicitly discuss as a natural direction for future work. These clarifications are incorporated into both the main text (§4.4) and Appendix A.1 to ensure that the paper’s scope and applicability are correctly understood.

In particular, we implemented the controlled comparison explicitly requested by Reviewer ajhU, constructing an optical-flow-based bias module under strictly identical Transformer backbone, training protocol, and bias-fusion scheme. Under these fully matched conditions, our Lie-rotation module achieves **34.06 dB**, outperforming the optical-flow-derived bias (**31.63 dB**) by **+2.43 dB**, directly validating our core claim that Lie-algebra rotation provides a stronger and more stable inductive bias than flow-based representations. We additionally added new attention-difference visualizations, controlled misalignment-correction examples, and high-resolution qualitative results. All controlled comparisons have been **integrated into the main ablation study**, with extended results included in **Appendix**.

Importantly, after examining these updates, **the previously skeptical reviewer (ajhU)**—who initially assigned a *score of 2*—confirmed that the new controlled experiment directly supports the paper’s central claim and subsequently **raised the score to 6**. Due to the recent OpenReview platform bug, this upward score change is not currently displayed; we hope to clearly communicate here that the reviewer acknowledged the revision and updated their evaluation accordingly.

We believe these comprehensive revisions, strengthened analyses, and the decisive controlled experiment fully address the reviewers’ concerns. Thank you very much for your time and consideration.

---

### Meta-Review · Area_Chair_z1hq · 2025-12-07

**Summary:**

This submission proposes DeLiVR, a video deraining method that injects spatiotemporal Lie-group differential biases (primarily SO(2) rotations) into Transformer attention to improve geometry-consistent alignment and temporal consistency without relying on optical flow. All three reviewers agree that the paper tackles an important problem in video restoration, introduces a conceptually novel and efficient use of Lie groups in attention, and demonstrates strong performance on synthetic and real rainy benchmarks (e.g., WeatherBench, NTU-Rain, Rain-Syn-Complex) with a compact model and good inference speed. The main concerns center on (i) the restriction to in-plane rotations (SO(2)) rather than full 3D motion, (ii) the strength of evidence that the Lie-group biases actually resolve cross-frame misalignment and temporal artifacts, (iii) the need for controlled comparisons against optical-flow-based modulation under identical backbones and fusion schemes, and (iv) some issues in presentation (figures, interpretation of LPIPS, clarity). After rebuttal and additional experiments—including a strictly controlled comparison between Lie-rotation biases and optical-flow-derived biases, extra visualization and stress-tests, and clarifications on the role and limitations of SO(2)—the reviewers converge to a mildly positive stance, with remaining concerns mostly framed as limitations rather than fatal flaws. Overall, I find the contributions sufficiently novel and validated for a poster recommendation.

**Reviewer Concerns:**

Concerns that were substantially addressed by the rebuttal and revisions

Controlled comparison vs optical flow (Reviewer ajhU, also related to others)

Initially, ajhU felt that the comparisons to optical-flow-based methods did not isolate whether the proposed rotation bias is actually stronger than optical flow guidance, since the models differed in more than one aspect.

The authors conducted the explicit controlled experiment requested by ajhU: same Transformer backbone, same training protocol, same bias-fusion locations, with a flow-based bias constructed from RAFT flow via a learnable MLP. Under these matched conditions on NTU-Rain, the Lie-rotation module (34.06 dB) outperforms the optical-flow-based bias (31.63 dB) by +2.43 dB. ajhU explicitly confirms that this directly supports the central claim that Lie-algebra rotation provides a stronger conditioning signal than optical flow, and raises their score accordingly. The authors also report that this comparison has been integrated into the main ablation study in the revised paper.

Does the method truly address cross-frame misalignment and temporal artifacts? (ajhU)

ajhU requested more interpretability and targeted evidence, not just aggregate metrics.

The authors added a controlled experiment where they inject small in-plane rotations into some frames and show new attention visualizations and attention-difference maps. According to the response, the rotation-aware model captures richer directional dependencies, and the attention maps selectively boost motion-sensitive structures, reducing flicker and ghosting. These results are now included in the main text’s ablation section. ajhU states that rebuttal clarifies most of their concerns (Q1, Q2, Q4).

Connection to deraining vs general video restoration; degradation-agnostic nature (ajhU, KGvX)

Reviewers questioned whether the method is really deraining-specific or more broadly a camera-motion prior.

The authors clarified that the predicted rotation corresponds to camera pose variation, not rain tilt, and that the geometric prior is degradation-agnostic. They report additional experiments on dehazing and deblurring (Appendix Fig. 7), which suggest the approach transfers to other degradations. This addresses the scope/applicability aspect, though the core deraining focus remains.

Role of Lie groups / stability vs non-rigid motion manifold (KGvX, HmPv)

KGvX and HmPv asked how the SO(2) Lie algebra parameterization relates to the true motion manifold and how it helps with stability and accuracy when motion is non-rigid.

The authors clarified that the Lie prior is not intended to model complex non-rigid rain and scene motion, but rather a low-dimensional rigid manifold (global in-plane camera rotation). The alignment is performed in the tangent space with an exponential map to guarantee valid rotations, which stabilizes optimization. This “peels off” camera jitter so that the Transformer can better focus on non-rigid rain dynamics. They also explain the hybrid attention formulation where data-driven terms can override misleading geometric bias in irregular motion or occlusions, alleviating concerns about over-reliance on geometry.

Presentation issues and figure inconsistencies (ajhU, KGvX, HmPv)

Reviewers noted rough figures and at least one inconsistency (Fig. 4) and some confusion in highlighting and metric interpretation.

The authors report revising Figure 4 to match the caption, adding high-resolution versions of Figure 3 in the appendix, and correcting the misinterpretation of LPIPS relative to VDMamba. They also acknowledge typos and presentational issues and state that they have refined the text accordingly. Reviewers do not raise further objections after rebuttal on these specific points.

Concerns that remain partially outstanding or are best viewed as limitations

Restriction to SO(2) and limited handling of full 3D camera motion (all reviewers, especially ajhU and HmPv)

All reviewers note that the current method focuses on in-plane rotation and may not fully handle general 3D camera motion (SE(3)) or extreme camera shake.

The authors clarify the motivation (small-angle in-plane rotations dominate in the datasets used, and SO(2) enables efficient, tractable bias computation) and emphasize that the formulation is mathematically extensible to SE(3). They also mention additional experiments exploring 3D camera motion scenarios and note some potential of their approach there.

However, the core method and main results remain SO(2)-based, and the paper does not fully resolve 3D motion in a comprehensive way. ajhU explicitly frames this as a limitation that can reasonably be left to future work; I agree that this should be clearly presented as such in the final version.

Breadth and depth of evaluation on real data and downstream tasks (KGvX, HmPv)

While the authors add more comparisons (including flow-based baselines) and some downstream evaluation on real rainy scenes with off-the-shelf detection, the bulk of quantitative evidence is still on synthetic benchmarks and a limited set of real evaluations.

Given venue constraints, this is acceptable, but it remains a point where the paper could be stronger, and it should be acknowledged as an experimental limitation.

Overall clarity and density of the mathematical exposition (mainly ajhU)

ajhU felt Section 3 was heavy on background formulas and light on clearly highlighted novel derivations and key ideas. The authors state they have clarified motivation and reorganized some explanations (e.g., in Sec. 4.3 and 4.4, Appendix A.1).

Without the full final PDF here, I cannot fully verify how far the exposition has improved, but given ajhU’s final, more positive stance, I infer that while some density remains, it is no longer a blocker, just an area where further polishing would still help.

**Reviewer Scores:**

Reviewer ajhU

Initially recommended rejection, mainly due to lack of convincing evidence that the Lie-group rotation bias truly outperforms optical-flow-based guidance and resolves cross-frame misalignment, as well as concerns about the SO(2) restriction and presentation.

After the rebuttal, especially after the controlled optical-flow vs Lie-rotation comparison under identical conditions and the added interpretability experiments, ajhU explicitly states that these results directly support the main claim and updates their overall judgment from negative to borderline positive, indicating support for acceptance provided that the missing 3D motion modeling is treated as a limitation/future work.

If ajhU had participated fully in the discussion with all final revisions in place, I expect they would maintain this borderline-accept stance, not revert to a negative recommendation.

Reviewer HmPv

From the beginning, HmPv viewed the paper as technically sound with clear contributions and good experiments, but with notable limitations (SO(2)-only, possible over-reliance on geometry, limited downstream evaluation).

The rebuttal addresses the geometric-prior role, stability rationale, and adds some downstream evidence and clarifications. HmPv’s written review already reflects a marginally positive recommendation and an overall favorable impression of the method’s novelty and rigor.

With the additional clarifications and controlled comparison requested by ajhU, I see no reason for HmPv to lower their evaluation; if anything, they might be slightly more confident in their mild-accept position.

Reviewer KGvX

KGvX is consistently positive about the core idea and experimental validation, while pointing out issues such as the incomplete connection to the true motion manifold, the earlier misinterpretation of LPIPS compared to VDMamba, and limited real-data downstream evaluation.

The authors respond by clarifying the role of the Lie prior, fixing the LPIPS discussion, and adding more details and links for additional responses.

Given that KGvX’s baseline stance is already marginally above the acceptance threshold, and the rebuttal addresses their main factual concerns, I expect KGvX would keep a mildly positive recommendation after full discussion.

---

### Decision · Program_Chairs · 2026-01-26

Accept (Poster)